# Fertility-Sparing Treatments in Endometrial Cancer: A Comprehensive Review on Efficacy, Oncological Outcomes, and Reproductive Potential

**DOI:** 10.3390/medicina61030471

**Published:** 2025-03-07

**Authors:** Carlo Ronsini, Paola Romeo, Giada Andreoli, Vittorio Palmara, Marco Palumbo, Giuseppe Caruso, Pasquale De Franciscis, Giuseppe Vizzielli, Stefano Restaino, Vito Chiantera, Stefano Cianci

**Affiliations:** 1Department of Woman, Child and General and Specialized Surgery, University of Campania “Luigi Vanvitelli”, 80138 Naples, Italy; carlo.ronsini@unicampania.it (C.R.); andreoli.giada@gmail.com (G.A.); pasquale.defranciscis@unicampania.it (P.D.F.); vito.chiantera@gmail.com (V.C.); 2Unit of Gynecology and Obstetrics, Department of Human Pathology of Adult and Childhood “G. Barresi”, University of Messina, 98121 Mesina, Italy; paolaromeo135@gmail.com (P.R.); vittorio.palmara@unime.it (V.P.); 3Department of General Surgery and Medical Surgical Specialties, University of Catania, 95124 Catania, Italy; mpalumbo@unict.it (M.P.); giu.caruso97@gmail.com (G.C.); 4Clinic of Obstetrics and Gynecology, “Santa Maria della Misericordia” University Hospital Azienda Sanitaria Universitaria Friuli Centrale, 33100 Udine, Italy; giuseppevizzielli@yahoo.it (G.V.); restaino.stefano@gmail.com (S.R.); 5Department of Medicine (DMED), University of Udine, 33100 Udine, Italy

**Keywords:** endometrial cancer, fertility-sparing, progesterone, levonorgestrel, intrauterine system

## Abstract

Endometrial cancer (EC) affects 3–14% of women under 40 who wish to preserve their fertility. The standard treatment for EC is a hysterectomy with salpingo-oophorectomy. However, for those desiring fertility preservation, oral progestogens such as medroxy-progesterone acetate (MPA) or megestrol acetate (MA) are the most common therapies in Fertility-Sparing Treatment (FST). Other treatments include gonadotropin-releasing hormone agonist (GnRHa), levonorgestrel-releasing intrauterine system (LNG-IUS), and metformin plus progestin. This comprehensive review evaluates the best FST options for women with reproductive potential. PubMed, EMBASE, and Scopus were searched in June 2023 using specific keywords. Studies included in the review focused on patients with EC undergoing FST, with outcomes such as complete response rate (CRR), recurrence rate (RR), pregnancy rate (PR), and live birth rate. Eighteen studies met the inclusion criteria, involving 23,976 patients. In only-oral progestin trials, CRR ranged from 18% to 100%; RR ranged from 0% to 81.8%; Death Rate ranged from 0% to 3.6%. In studies combining oral progestin with LNG-IUS, CRR ranged from 55% to 87.5%; RR ranged from 0% to 41.7%; Death Rate was 0%. Most patients with Stage IA EC received MPA or MA. Fertility-related outcomes were reported in 15 studies. PR ranged from 4 to 44 patients in trials involving only oral progestins. When combining oral progestin with LNG-IUS, PR ranged from 1 to 46 patients. Progestin therapy, including oral MPA and MA, is considered safe and effective, with limited evidence supporting the use of LNG-IUS.

## 1. Introduction

Endometrial cancer is one of the most common malignancies of the female reproductive system worldwide [1]. While EC primarily affects women over 50, following menopause, about 25% of cases occur before menopause [2]. In particular, 3–14% of EC cases are diagnosed in women under 40 who wish to preserve their fertility [3,4]. The exact cause of EC remains unclear, but several risk factors are known. EC is estrogen-dependent, and factors such as obesity, which increases estrogen production through aromatization in adipose tissue, hyperinsulinemia, diabetes, hypertension, nulliparity, anovulatory cycles, and a sedentary lifestyle elevate the risk [5,6]. EC has been historically classified into two types: type I accounts for 85% of cases, is estrogen-dependent, and generally less aggressive; type II, which includes more aggressive forms such as serous, clear cell, and undifferentiated carcinomas, is often associated with older age, higher stage, and poorer prognosis [5,7]. In 2013, the Cancer Genome Atlas (TCGA) endometrial collaborative project further classified EC into four distinct prognostic subtypes based on genomic abnormalities: Deoxyribonucleic Acid (DNA) polymerase epsilon (POLE) gene ultramutated, microsatellite instability (MSI) hypermutated, copy number low, and copy number high [8]. Subsequently, Talhouk et al. [9] developed a simplified molecular classification of EC, called the Proactive Molecular Risk Classifier for Endometrial Cancer (ProMisE), to improve risk stratification and, consequently, the management of patients with endometrial cancer. In this classification, unlike TCGA, MSI assessment and Tumor Protein P53 (TP53) sequencing are obtained by immunochemistry surrogates: mismatch repair (MMR) proteins and p53, respectively. In particular, current prognoses are based on molecular features, as per the 2023 FIGO (International Federation of Gynecology and Obstetrics) Classification [10]. For EC treatment, hysterectomy with salpingo-oophorectomy is the standard [11]. However, younger women seeking fertility preservation may opt for conservative treatments. Oral progestogens—medroxyprogesterone acetate (MPA) or megestrol acetate (MA)—are the most commonly used therapies in Fertility-Sparing Treatment (FST) [12]. Other treatments include gonadotropin-releasing hormone agonist (GnRHa), levonorgestrel-releasing intrauterine system (LNG-IUS), and metformin plus progestin [12]. Although international guidelines endorse fertility-sparing treatments for young cancer patients [13], they must be informed of the need for close follow-up due to the risk of cancer recurrence or persistence [14]. This systematic review evaluated whether fertility-sparing treatments in EC are safe and effective for young women with reproductive potential and which regimen is most appropriate for this population.

## 2. Materials and Methods

We searched for articles in the PubMed, Embase, and Scopus databases in June 2023, adopting the string (“Endometrial Neoplasms” [Mesh]) AND “Fertility Preservation” [Mesh]) AND “Levonorgestrel” [Mesh]. MeSH terms were used to find relevant literature by grouping related topics under consistent headings, regardless of the exact words used. We considered English-published articles with no restrictions on country or year of publication. Study selection was conducted independently by I.I. and E.B. In case of discrepancy, C.R. decided on inclusion or exclusion. Inclusion criteria were (1) studies that included patients with a histological diagnosis of EC; (2) studies with patients undergoing FST; (3) articles reporting at least one outcome of interest—Complete Response Rate (CRR), Recurrence Rate (RR), Death Rate, percentage of women attempting to conceive, Pregnancy Rate (PR), Live Birth Rate, and Preterm Rate; (4) peer-reviewed articles published originally. We excluded non-original studies, preclinical trials, animal trials, abstract-only publications, and articles in languages other than English. We assessed all included studies for potential conflicts of interest.

## 3. Results

### 3.1. Studies Characterisctics

Fifty articles matched the search criteria. A total of 26 records were eligible after removing records with no full text, duplicates, or incorrect study designs (e.g., reviews). Of those, 18 matched the inclusion criteria and were included in the review. Ten articles were prospective cohort studies evaluating FST in patients with EC [15,16,17,18,19,20,21,22,23,24]; eight records were prospective cohort studies [25,26,27,28,29,30,31,32]. The countries where the studies were conducted, the publication year range, the studies’ design, number of participants, FIGO (International Federation of Gynecology and Obstetrics) stage, type of FST, and months of follow-up (FU) are summarized in Table 1. The publication years ranged from 2009 to 2021. A total of 23,976 patients with EC were included, and the FIGO stage of disease ranged from I to II. Median FU months were 6.0–98.0 [15,16,17,18,19,20,21,22,23,24,25,26,27,28,29,30,31,32].

### 3.2. Outcomes

#### 3.2.1. Oncological Outcomes

##### Oncological Outcomes in Only Oral-Progestin Studies

In studies involving only oral progestins, a total of 23,633 patients were treated. The CRR ranged from 18% to 100%; the RR ranged from 0% to 81.8%; the Death Rate ranged from 0% to 3.6%. Patients’ median age ranged from 30 to 35 years old. FU time ranged from 6 to 98 months. All studies included patients with FIGO Stage I disease, while one study also included patients with Stage II. The period of enrollment ranged from 3 years up to 15 years. Data are summarized in Table 2.

##### Oncological Outcomes in Oral Progestin and LNG-IUS Combination Studies

In trials combining oral progestin with LNG-IUS, 312 patients were treated. The CRR ranged from 55% to 87.5%; the RR ranged from 0% to 41.7%; the Death Rate was 0% in all trials. FU time ranged from 6 to 61 months. Patients’ median age ranged from 30.4 to 38.5 years old. All studies included patients with FIGO Stage I of the disease, four included those with Stage II. The period of enrollment ranged from 2 to 14 years. Those data are summarized in Table 3.

#### 3.2.2. Fertility Outcomes

In total, 15 studies over 18 presented pregnancy outcomes [14,15,16,17,19,20,21,23,26,27,28,29,30].

#### Fertility Outcomes in Only Oral-Progestin Studies

In only-oral progestin studies, a total of 330 patients were evaluated. Patients’ median age ranged from 30 to 35 years. The median FU ranged from 38 to 91 months; the PR ranged from 4 to 44 patients. All studies included patients with FIGO Stage I disease, while one study also included patients with Stage II. The period of enrollment ranged from 3 to 15 years. Data are summarized in Table 4.

#### Fertility Outcomes in Oral Progestin and LNG-IUS Combination Studies

In trials combining oral progestin with LNG-IUS, a total of 299 patients were evaluated. Patients’ median age ranged from 32.8 to 38.5 years. The median FU ranged from 6 to 44 months; the Pregnancy Rate ranged from 1 to 46 patients. All studies included patients with FIGO Stage I disease, while four studies also included patients with Stage II. The period of enrollment ranged from 2 years to 14 years. Data are summarized in Table 5.

## 4. Discussion

EC is one of the most common malignancies of the female reproductive system. About 14% of cases occur before menopause in women who may wish to preserve their fertility [1].

This comprehensive review evaluates whether fertility-sparing treatments in EC are safe and effective for young women with reproductive potential and offers valuable insights into the various FST options for EC—an issue that has been scarcely investigated. However, the main limitation of the review is that the scarcity and heterogeneity of available data limited our analysis.

The main findings of this study suggest that tumor grading significantly impacts oncological outcomes. Previous research indicates that the complete response rate (CRR) is generally lower in patients with grade 2 (G2) endometrial cancer (EC) compared to those with grade 1 (G1) EC, as G1 tumors tend to be more responsive to progesterone therapy [16,25,27,33]. Our analysis confirms that patients treated with medroxyprogesterone acetate (MPA) and megestrol acetate (MA) exhibited the highest CRR and lowest recurrence rates (RR), particularly in studies involving G1 EC, further supporting the idea that G1 EC is more progesterone-sensitive [16,25,28]. However, the influence of tumor differentiation and grading on pregnancy outcomes remains unclear. Indeed, as reported in previous literature, patients with G1 and G2 EC can both have higher levels of Plasminogen Activator Inhibitor-1 (PAI-1), a regulator of fibrinolysis, which may contribute to poor fertility outcomes despite fertility-sparing treatment (FST) [34,35,36]. In fact, elevated levels of PAI-1 in EC patients inhibit fibrinolysis, which may lead to thrombosis in placental and uterine vessels, impairing uterine function and fertility. Moreover, the prothrombotic environment and impaired tissue remodeling could decrease endometrial receptivity, reducing the likelihood of embryo implantation [37]. In addition, high PAI-1 levels are also associated with poor oncological prognoses, indicating a more aggressive phenotype [34]. Therefore, careful counseling is essential in these cases, as fertility-sparing treatments’ aim should be a balance between oncological control and preservation of reproductive function. However, the risk of inadequate disease control may outweigh the lower chances of achieving a successful pregnancy, thus making individualized decision-making crucial. It would have been interesting to analyze the need for Assisted Reproductive Technology (ART) to achieve conception and the subsequent live birth or miscarriage. Unfortunately, this is a limitation of the present study since those data were not available in the selected records. Hence, this aspect should be investigated in further prospective studies. Only Chae et al. show that 44.9% of their cohort conceived, 23.3% ended up in miscarriages, and 3.3% in preterm birth. Live birth rate was 66.6% [26].

Concerning additional elements that could ameliorate treatment efficacy, hysteroscopic endometrial resection and a long duration of treatment (at least 6 months) can be considered [28,38,39,40,41]. Indeed, treatment strategies such as hysteroscopic endometrial resection combined with high-dose progesterone (either MPA at a dose of 400–600 mg/d or MA at a dose of 160–320 mg/d) have shown a 100% CRR [28,41]. However, in this case, it is also important to carefully select patients based on tumor grade, stage, and individual risk factors. To ensure effectiveness and safety, hysteroscopic resection combined with progesterone therapy is usually reserved for patients with early-stage, low-grade endometrial cancer or atypical hyperplasia. Moreover, it is important to note that regular follow-up hysteroscopies and biopsies are essential to detect any possible recurrence or progression of the disease. It is clear that standardized treatment protocols, covering the optimal dosage of oral MPA and MA, treatment duration, and follow-up, are crucial for improving patient outcomes. While a clear consensus is yet to be established, the literature commonly reports MPA and MA dosage ranges of 200–800 and 40–400 mg/day, respectively. Current recommendations suggest MPA at a dose of 400–600 mg/day or MA at a dose of 160–320 mg/day [3,12]. Treatments usually last 6–12 months, with regular monitoring through endometrial biopsy or hysteroscopy every 3–6 months to assess the response to therapy [28,30,40,41,42].

This regimen could be at the expense of the patient, who must undergo intensive monitoring. However, this approach has shown promising results not only in terms of CRR but also in fertility. Indeed, a study demonstrated that in six out of twenty-three patients treated, seven pregnancies were achieved after an average time of 7.4 months (range 3–13 months) after the end of progestin therapy [41].

Alternatives to MPA and MA that could be considered are gonadotropin-releasing hormone agonists (GnRHa) alone or with a levonorgestrel-releasing intrauterine system (LNG-IUS) [39,42]. As reported in the literature, this treatment is reserved for grade 1 endometrial cancer or atypical hyperplasia. In particular, those who received GnRHa alone had first undergone endometrial resection and laparoscopy to exclude concomitant ovarian tumor and/or other extra-uterine disease. They also underwent a follow-up hysteroscopy every 3 months to exclude recurrence or progression. CRR was 67% [39]. On the other hand, Pronin et al. treated patients with atypical endometrial hyperplasia with LNG-IUS and those with grade 1 endometrial cancer with LNG-IUS combined with GnRHa, showing complete remission in 92% of patients with atypical hyperplasia and 72% of those with grade 1 EC [42].

Another important point of discussion is metformin, which regulates glucose metabolism. It has shown promise in enhancing progestin response by increasing progesterone receptor expression, with studies reporting a CRR of 80% in G1 EC patients [43]. In particular, Zhou et al. demonstrated that EC patients with elevated HbA1C had higher CRR when treated with metformin and progesterone [44]. While metformin’s role in fertility remains unclear due to limited data, it may also support weight control, further improving fertility outcomes [45]. Unfortunately, neither of the studies included in our review revealed pregnancy outcomes in EC patients administered with metformin.

Regarding the risk of recurrence, it seems that women who conceive after treatment tend to experience later cancer recurrence compared to those who do not [26]. In particular, factors such as preconception relapse, endometrial thickness, and age may play a role in recurrence risk [46]. Although hormonal treatment duration does not appear to affect pregnancy outcomes in patients undergoing FST for EC, there is no standardized recommendation for the length of therapy [47,48,49,50,51,52,53]. This lack of consensus may be due to the variability in patient responses, which results from individual differences and tumor characteristics, as well as the absence of large-scale studies comparing different treatment durations. For this reason, treatment duration is often individualized depending on patient preferences, tumor grade, and response to therapy, which presents another limitation of our analysis. Indeed, there is a lack of information about the intercurrent time between the start of treatment and response. Moreover, the reported median time from treatment start to histologic response shows wide variations in the scientific literature. Patients often undergo multiple endometrial sampling every three months. Due to the retrospective design of the analysis, it was not possible to assess whether some studies treated their cohort for more than the standard duration of 6 months. In the 2012 study by Park et al., patients underwent treatment for durations ranging from 3 to 15 months [16]. In their 2013 analysis, the mean duration of progestin administration varied from 2 to 31 months [18]. Mitsuhashi et al. treated their cohort with MPA and metformin for 24– 36 months to obtain CR [20]. Similarly, Hwang et al. treated patients from 6 to 18 months [23], while Chae et al. reported treatment durations spanning from 3 to 18 months [29]. Collectively, these data may also reflect cohorts comprising both endometrial cancer (EC) and atypical endometrial hyperplasia.

Another issue to consider when evaluating the possibility of a patient undergoing FST for EC is comorbidities. In particular, risk factors—as PCOS, metabolic syndrome, and obesity—may be obstacles for reproductive potential. Hence, it would be appropriate to organize a multidisciplinary setting of care that embraces all aspects of the disease.

### Impact of New FIGO Staging

The updated 2023 FIGO staging included various histological types, tumor patterns, and molecular classification [10]. It has been shown that the different molecular markers have changed the risk profiles. Not all stage IA grade 1 tumors present the same biology; therefore, they have different associated outcomes [54]. Recent studies have explored whether molecular classifiers play a role in the selection of candidates for fertility-sparing treatment [55]. In particular, Puechl et al. retrospectively found that among patients who underwent FST, those with p53 abnormal tumors had a shorter time to progression or definitive therapy than those with POLE-mutated and MMR-deficient tumors (less than 6 months, compared to around 20 months, respectively) [56]. Similarly, Zhang et al. showed that most patients considered eligible for fertility-sparing management had p53 wild-type tumors, whereas p53 abnormal tumors were associated with worse prognoses [57]. Additionally, Falcone et al. explored the potential role of molecular classification in the selection of patients for FST. They demonstrated that three out of seven patients with MMR deficiency had disease persistence, progression, and/or relapse, whereas two out of eight patients with no mutation or POLE mutations had a relapse of their disease [58]. These results demonstrate that the new FIGO staging system might play a fundamental role in the better selection of patients and tailored fertility-sparing treatment. In particular, we might infer from the existing literature that POLE-mutated and MMR-deficient tumors are more likely to respond to hormonal therapy, thereby benefiting from fertility-sparing treatment. Conversely, p53 abnormal subtypes are more aggressive, making them less suitable for fertility-sparing approaches due to their higher risk of progression. However, further prospective studies are needed in this field.

An important limitation of our study is the lack of information regarding ESGO risk classification [59]. Unfortunately, the available data in selected articles did not allow us to perform an accurate stratification that could be assessed in further prospective studies.

## 5. Conclusions

In conclusion, FST in young patients with early-stage, low-grade EC is feasible and safe. Progestin therapy is the most valid option in terms of both CR and PR. Hysteroscopic resection prior to progestin therapy could ameliorate treatment efficacy. Research regarding the use of LNG-IUS either alone or in combination with progestins is lacking, and further studies are needed. Moreover, a clear consensus on the regularity of hysteroscopic follow-up after FST for EC and on the length of hormonal treatment before starting to look for a pregnancy is needed. Individualized decision-making is always crucial and further randomized clinical trials are necessary to test the effectiveness of other conservative regimens. Important aspects to consider for future perspectives are an accurate stratification of patients according to ESGO risk classification and an assessment of live birth rates among pregnant patients.

## Figures and Tables

**Table 1 medicina-61-00471-t001:** Characteristics of included studies.

Author(s), Year	Country	Study Design	Period of Enrollment	Median Age (Years)	FIGO Stage	No.	FST	Median FU (Months)
Signorelli, 2009 [15] ^	Italy	Prospective monocenter cohort study	1992–2004	32.0	IA	11	Oral MPA/GnRHa/Danazole	98.0
Park, 2012 [16]	Korea	Retrospective monocenter cohort study	N/A	30.0	I-II	14	Oral MPA/MA	47.3
Kim, 2013 [17]	Korea	Prospective monocenter cohort study	2008–2012	34.8	I-II	16	Oral MPA and LNG-IUS	31.1
Park, 2013 [18]	Korea	Prospective multicenter cohort study	1996–2010	31.3	IA	148	Oral MPA/MA	66.0
Kudesia, 2014 [19] ^	USA	Prospective monocenter cohort study	2000–2011	38.5	I	10	Oral progesterone and/or LNG-IUS	13.0
Mitsuhashi, 2015 [20] ^	Japan	Prospective monocenter cohort study	2009–2012	33.0	IA	19	MPA and metformin	38.0
Ohyagi-Hara, 2015 [21] ^	Japan	Prospective monocenter cohort study	2000–2011	34.2	IA	16	Oral MPA	39.2
Chen, 2016 [22] ^	China	Retrospective monocenter cohort	2000–2011	32.0	IA	53	Oral MPA/MA	6.0
Hwang, 2017 [23]	Korea	Retrospective monocenter cohort study	2011–2015	30.4	IA	5	Oral MPA and LNG-IUS	44.4
Ruiz, 2017 [24]	USA	Retrospective multicenter cohort study	2004–2014	N/A	I	23231	Oral progesterone	54.0
Yamagami, 2018 [25] ^	Japan	Prospective monocenter cohort study	1998–2013	35.0	IA	97	Oral MPA	71.3
Chae, 2019 [26]	Korea	Retrospective multicenter cohort study	2005–2017	37.0	IA	118	Oral MPA and LNG-IUS	9.0
Kim, 2019 [27]	Korea	Prospective multicenter cohort study	2012–2017	32.9	I-II	44	Oral MPA and LNG-IUS	6.0
Yang, 2019 [28]	Taiwan	Prospective monocenter cohort study	2013–2017	33.6	IA	6	Oral MPA/MA	32.0
He, 2020 [29] ^	China	Retrospective monocenter cohort study	2005–2019	32.8	IA	16	Oral MPA/MA or in combination with LNG-IUS/GnRHa	61.0
Xu, 2020 [30] ^	China	Prospective monocenter case-control study	2014–2016	33.3	I-II	96	LNG IUS and/or high-efficient MA	N/A
İşçi Bostancı, 2021 [31]	Turkey	Retrospective monocenter cohort study	2005–2020	34.8	IA-IC-IIIC2	38	Oral MPA/MA	40.5
Piatek, 2021 [32] ^	Poland	Retrospective multicenter cohort study	2010–2019	30.6	I-II	38	Intramuscular MPA/Oral MA/LNG-IUS	36.5

^ sub-analysis of entire cohort. FIGO: International Federation of Gynecology and Obstetrics; FST: fertility-sparing treatment; FU: follow-up; MPA: medroxyprogesterone acetate; MA: megestrol acetate; LNG-IUS: levonorgestrel intrauterine system; GnRHa: gonadotropin releasing hormone agonist; N/A: not available.

**Table 2 medicina-61-00471-t002:** Oncological outcomes for single-oral agents.

Author(s), Year	FST	Histological Subtype	Grading	FIGO Stage	Recurrence Rate (%)	Death Rate (%)	Complete Response (%)
Signorelli, 2009 [15] ^	Oral MPA/GnRHa/Danazole	N/A	1	IA	N/A	0.00	18.0
Park, 2012 [16]	Oral MPA/MA	Endometrioid	1	I-II	31.0	0.00	92.8
Park, 2013 [18]	Oral MPA/MA	Endometrioid	1	IA	32.0	0.00	77.7
Mitsuhashi, 2015 [20] ^	MPA and metformin	Endometrioid	1	IA	20.0	0.00	80.0
Ohyagi-Hara, 2015 [21] ^	Oral MPA	Endometrioid	1	IA	81.8	0.00	68.8
Chen, 2016 [22] ^	Oral MPA/MA	Endometrioid	1	IA	22.0	0.00	73.0
Ruiz, 2017 [24]	Oral progesterone	Endometrioid	1–2–3	I	40.0	3.6	55.0
Yamagami, 2018 [25] ^	Oral MPA	Endometrioid	1	IA	63.2	0.00	90.7
Yang, 2019 [28]	MPA/MA	Endometrioid	1	IA	0.00	0.00	100
İşçi Bostancı, 2021 [31]	Oral MPA/MA	Endometrioid	1–2	IA-IC-IIIC2	78.9	0.00	30.3

^ sub-analysis of entire cohort; FST: fertility-sparing treatment; FIGO: International of Gynecology and Obstetrics; N/A: not available; MPA: medroxyprogesterone acetate; MA: megestrol acetate; LNG-IUS: levonorgestrel intrauterine system; GnRHa: gonadotropin releasing hormone agonist.

**Table 3 medicina-61-00471-t003:** Oncological outcomes for combined therapy.

Authors, Year	FST	Histological Subtype	Grading	FIGO Stage	Recurrence Rate (%)	Death Rate (%)	Complete Response (%)
Kim, 2013 [17]	Oral MPA and LNG-IUS	Endometrioid	1	I-II	0.00	0.00	87.5
Kudesia, 2014 [19] ^	Oral progesterone and/or LNG-IUS	Endometrioid	1	I	N/A	0.00	70.0
Hwang, 2017 [23]	Oral MPA and LNG-IUS	Endometrioid	2	IA	20.0	0.00	60.0
Chae, 2019 [26]	MPA and LNG-IUS	Endometrioid	1–2	IA	36.7	0.00	60.1
Kim, 2019 [27]	MPA and LNG-IUS	Endometrioid	1	I-II	14.3	0.00	87.5
He, 2020 [29] ^	Oral MPA/MA or in combination with LNG IUS/GnRHa	Endometrioid	1–2	IA	41.7	0.00	75.0
Xu, 2020 [30] ^	LNG IUS and/or high-efficient MA	Endometrioid	N/A	I-II	13.3	0.00	86.5
Piatek, 2021 [32] ^	Intramuscular MPA/Oral MA/LNG-IUS	Serous, endometrioid	1–2	I-II	15.0	0.00	55.0

^ sub-analysis of entire cohort; FST: fertility-sparing treatment; FIGO: International of Gynecology and Obstetrics; N/A: not available; MPA: medroxyprogesterone acetate; MA: megestrol acetate; LNG-IUS: levonorgestrel intrauterine system; GnRHa: gonadotropin releasing hormone agonist.

**Table 4 medicina-61-00471-t004:** Fertility outcomes for single-oral agent.

Authors, Year	FST	Attempted to Conceive/All Patients (%)	Pregnancy Rate (No.)	Birth Rate (No.)	Preterm Rate (No.)	Median FU (Months)
Signorelli, 2009 [15] ^	Oral MPA/GnRHa/Danazole	100	4	N/A	N/A	98.0
Park, 2012 [16]	Oral MPA/MA	50.0	4	4	3	47.3
Park, 2013 [18]	Oral MPA/MA	38.3	44	44	N/A	66.0
Mitsuhashi, 2015 [20] ^	MPA and metformin	N/A	N/A	N/A	N/A	38.0
Ohyagi-Hara, 2015 [21] ^	Oral MPA	100	1	1	N/A	39.2
Ruiz, 2017 [24]	Oral progesterone	N/A	N/A	N/A	N/A	54.0
Yamagami, 2018 [25] ^	Oral MPA	90.0	20	N/A	N/A	71.3
Yang, 2019 [28]	MPA/MA	N/A	6	1	0	32.0
İşçi Bostancı, 2021 [31]	Oral MPA/MA	84.2	7	5	N/A	40.5

^ sub-analysis of entire cohort. FST: fertility-sparing treatment; FU: follow-up; N/A: not available; MPA: medroxyprogesterone acetate; MA: megestrol acetate; LNG-IUS: levonorgestrel intrauterine system; GnRHa: gonadotropin-releasing hormone agonist.

**Table 5 medicina-61-00471-t005:** Fertility outcomes for combined therapy.

Authors, Year	FST	Attempted to Conceive/All Patients (%)	Pregnancy Rate (No.)	Birth Rate (No.)	Preterm Rate (No.)	Median FU (Months)
Kim, 2013 [17]	Oral MPA and LNG-IUS	56.2	3	2	N/A	31.1
Kudesia, 2014 [19] ^	Oral progesterone and/or LNG-IUS	100	10	2	N/A	13.0
Hwang, 2017 [23]	Oral MPA and LNG-IUS	40.0	1	1	0.00	44.4
Chae, 2019 [26]	MPA and LNG-IUS	41.5	22	20	1	9.0
Kim, 2019 [27]	MPA and LNG-IUS	N/A	N/A	N/A	N/A	6.0
He, 2020 [29] ^	Oral MPA/MA or in combination with LNG-IUS/GnRHa	N/A	N/A	3	0	19.5
Xu, 2020 [30] ^	LNG-IUS and/or high-efficient MA	76.0	46	21	N/A	N/A
Piatek, 2021 [32] ^	Intramuscular MPA/Oral MA/LNG-IUS	100	3	2	N/A	36.5

^ sub-analysis of entire cohort. FST: fertility-sparing treatment; FU: follow-up; N/A: not available; MPA: medroxyprogesterone acetate; MA: megestrol acetate; LNG-IUS: levonorgestrel intrauterine system; GnRHa: gonadotropin-releasing hormone agonist.

## Data Availability

No new data were created or analyzed in this study.

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
