# Peer review of "Fertility-Sparing Treatments in Endometrial Cancer: A Comprehensive Review on Efficacy, Oncological Outcomes, and Reproductive Potential"

_medicina, 2025, doi:10.3390/medicina61030471_

Round 1
Reviewer 1 Report
Comments and Suggestions for Authors
I would like to congratulate the authors for such an interesting review that approached a scarcely documented pathology: Fertility sparing treatments in women with endometrial cancer. They documented the efficacy of different types of treatments (oral progestin MPA/MA; oral progestin combined with LNG-IUS; GnRH alone or with LNG-IUS or metformin). Although in need for more randomized research the results are very interesting and might constitute a starting point for future more organized research.
Abstract section, Introduction and discussion section – please state the exact treatments and combinations from the included research and please use the same abbreviation for LNG-IUS for better understanding (not IUS and afterwards LNG-IUS).
Abstract section the last paragraph – “combining oral progestin” – please explain – association between MA and MPA? Both of them administered in the same patients.
Introduction section:
Row 43 – please correct “25%-14%” and “3%-14% around 5% of EC cases”
Please explain the abbreviation DNA, TP53, FIGO (explain it when it is mentioned for the first time in the article)
Materials and methods
What Mesh mean?
Results section
Please explain the abbreviation – FU
In 3.2.1 please divide the paragraph into 2 for more clarity and better understanding; the same comment in 3.2.2
Row 106 – “enrollment ranged from 15-3 years” – please correct into an ascending trend
Correct IUS with LNG-IUS and be more specific when describing the treatment associations; perhaps the dose of MPA and MA will help the reader
Discussion section
Row 197 – please try to be more specific – what does long duration treatment mean? And high dose progesterone?
Author Response
Reviewer 1
I would like to congratulate the authors for such an interesting review that approached a scarcely documented pathology: Fertility sparing treatments in women with endometrial cancer. They documented the efficacy of different types of treatments (oral progestin MPA/MA; oral progestin combined with LNG-IUS; GnRH alone or with LNG-IUS or metformin). Although in need for more randomized research the results are very interesting and might constitute a starting point for future more organized research.
Abstract section, Introduction and discussion section – please state the exact treatments and combinations from the included research and please use the same abbreviation for LNG-IUS for better understanding (not IUS and afterwards LNG-IUS).
-We thank the reviewer for the comment. The issue was addressed, see lines 21-26.
Abstract section the last paragraph – “combining oral progestin” – please explain – association between MA and MPA? Both of them administered in the same patients.
-We thank the reviewer for pointing this out. The combination is between oral progestin and LNG-IUS. We specified this better in the last paragraph of the abstract. See line 35.
Introduction section:
Row 43 – please correct “25%-14%” and “3%-14% around 5% of EC cases”
-We apologize for the mistake. Corrections were made. See lines 49-50.
Please explain the abbreviation DNA, TP53, FIGO (explain it when it is mentioned for the first time in the article)
-The abbreviations were explained. Please see lines 60,66,68,69.
Materials and methods
What Mesh mean?
-We thank the reviewer for the question, we specified this better in the methods section. MeSH stands for Medical Subject Headings. MeSH terms are standardized keywords that help researchers find relevant literature efficiently by grouping related topics under consistent headings, regardless of the exact words used by different authors. See lines 84,85.
Results section
Please explain the abbreviation – FU
-The abbreviation was explained. Please see line 104,105
In 3.2.1 please divide the paragraph into 2 for more clarity and better understanding; the same comment in 3.2.2
-Paragraphs were further divided according to the reviewer’s suggestion. Please see lines 126 and 132.
Row 106 – “enrollment ranged from 15-3 years” – please correct into an ascending trend
-Thank you for the suggestion. This was corrected, please see line 115.
Correct IUS with LNG-IUS and be more specific when describing the treatment associations; perhaps the dose of MPA and MA will help the reader
- We replaced IUS with LNG-IUS. As for treatment dosages, they were not specifically analyzed since most of the included studies follow standardized protocols. To ensure clarity, we have added this explanation in the discussion section. Please refer to lines 223,224
Discussion section
Row 197 – please try to be more specific – what does long duration treatment mean? And high dose progesterone?
-Long duration of treatment means at least 6 months, whereas high dose progesterone means either MPA at a dose of 400-600 mg/d or MA at a dose of 160-320 mg/d for at least 6 months. This was better clarified in lines 210 and 212.
Reviewer 2 Report
Comments and Suggestions for Authors
This comprehensive review evaluates fertility-sparing treatments (FST) in endometrial cancer (EC), focusing on oncological outcomes and reproductive potential. The authors systematically reviewed 18 studies involving 23,976 patients, analyzing complete response rates (CRR), recurrence rates (RR), pregnancy rates (PR), and live birth rates. The review provides valuable insights but has some limitations, including heterogeneity in study designs, lack of standardized treatment protocols, and incomplete data on long-term reproductive outcomes.
The manuscript presents an important and timely review of FST in EC, addressing both oncological safety and reproductive outcomes. The methodology is rigorous, and the discussion highlights critical factors influencing treatment efficacy.
However, there are some areas for improvement: the review includes studies with different FIGO stages, histological subtypes, and treatment durations, leading to substantial heterogeneity. While this is addressed in the discussion, a subgroup analysis (e.g., comparing outcomes in Stage IA vs. Stage II) would improve clarity. In addition, the duration of progestin therapy varies significantly (from 6 to 98 months), which may contribute to differences in CRR and RR. Were any efforts made to standardize the treatment duration across included studies? Moreover, the review evaluates both oral progestin monotherapy and combined approaches (e.g., MPA + LNG-IUS), but the discussion does not clearly define which approach is preferable based on patient characteristics.
The pregnancy and live birth rates are reported but lack information on the use of assisted reproductive technology (ART). Given the high recurrence rates, ART might be critical for achieving pregnancy before disease progression. A discussion on ART's role in fertility preservation would be beneficial. Also the review does not address long-term reproductive health (e.g., miscarriage rates, pregnancy complications, ovarian function post-treatment). These factors should be considered when counseling patients. It should be merntioned, that the discussion on the 2023 FIGO classification and molecular subtypes (POLE-mutated, MMR-deficient, p53-abnormal) is valuable. However, how should these molecular profiles influence FST selection? Would patients with p53-abnormal EC be unsuitable candidates for conservative management? The authors mention that molecular profiling is underutilized in clinical decision-making. Could they propose a standardized molecular-based algorithm for FST eligibility?
This review provides an in-depth analysis of FST in EC, highlighting key oncological and reproductive outcomes. However, several limitations should be addressed:
- Standardization of treatment protocols: More discussion on the ideal duration of progestin therapy and follow-up is needed.
- Molecular risk stratification: Further recommendations on how genomic classification should influence FST selection.
- Reproductive outcomes and ART: Addressing the role of fertility preservation techniques, including ART, would provide a more holistic perspective.
- Long-term follow-up recommendations: Guidance on post-treatment surveillance and recurrence monitoring is crucial.
Author Response
This comprehensive review evaluates fertility-sparing treatments (FST) in endometrial cancer (EC), focusing on oncological outcomes and reproductive potential. The authors systematically reviewed 18 studies involving 23,976 patients, analyzing complete response rates (CRR), recurrence rates (RR), pregnancy rates (PR), and live birth rates. The review provides valuable insights but has some limitations, including heterogeneity in study designs, lack of standardized treatment protocols, and incomplete data on long-term reproductive outcomes.
The manuscript presents an important and timely review of FST in EC, addressing both oncological safety and reproductive outcomes. The methodology is rigorous, and the discussion highlights critical factors influencing treatment efficacy.
However, there are some areas for improvement: the review includes studies with different FIGO stages, histological subtypes, and treatment durations, leading to substantial heterogeneity. While this is addressed in the discussion, a subgroup analysis (e.g., comparing outcomes in Stage IA vs. Stage II) would improve clarity. In addition, the duration of progestin therapy varies significantly (from 6 to 98 months), which may contribute to differences in CRR and RR. Were any efforts made to standardize the treatment duration across included studies? Moreover, the review evaluates both oral progestin monotherapy and combined approaches (e.g., MPA + LNG-IUS), but the discussion does not clearly define which approach is preferable based on patient characteristics.
The pregnancy and live birth rates are reported but lack information on the use of assisted reproductive technology (ART). Given the high recurrence rates, ART might be critical for achieving pregnancy before disease progression. A discussion on ART's role in fertility preservation would be beneficial. Also the review does not address long-term reproductive health (e.g., miscarriage rates, pregnancy complications, ovarian function post-treatment). These factors should be considered when counseling patients. It should be mentioned, that the discussion on the 2023 FIGO classification and molecular subtypes (POLE-mutated, MMR-deficient, p53-abnormal) is valuable. However, how should these molecular profiles influence FST selection? Would patients with p53-abnormal EC be unsuitable candidates for conservative management? The authors mention that molecular profiling is underutilized in clinical decision-making. Could they propose a standardized molecular-based algorithm for FST eligibility?
This review provides an in-depth analysis of FST in EC, highlighting key oncological and reproductive outcomes. However, several limitations should be addressed:
- Standardization of treatment protocols: More discussion on the ideal duration of progestin therapy and follow-up is needed.
- Thank you for your feedback. We agree that the standardization of treatment protocols is essential for improving patient outcomes. Unfortunately, there is no clear consensus yet, but we have expanded the discussion in the manuscript to provide clearer guidance. Please see lines 219-223.
- Molecular risk stratification: Further recommendations on how genomic classification should influence FST selection.
-We thank the reviewer for the suggestion. This was addressed in lines 295-300.
- Reproductive outcomes and ART: Addressing the role of fertility preservation techniques, including ART, would provide a more holistic perspective.
-We thank the reviewer for the suggestion. As mentioned in the limitations of our study, it would have been interesting to analyze the need of Assisted Reproductive Technology (ART) to achieve con-ception, and the subsequent live birth or miscarriage. Unfortunately, those data were not available in selected records, hence, that aspect should be investigated in further prospective studies. We made clearer that this is a limitation of the study (line 204,205).
- Long-term follow-up recommendations: Guidance on post-treatment surveillance and recurrence monitoring is crucial.
-Commonly used follow-up regimens include hysteroscopy and endometrial biopsy every 3-6 months for the first two years. We tried to make this clearer in the discussion. Please see lines 219-223.